# Breeding Enhancement of *Musca domestica* L. 1758: Egg Load as a Measure of Optimal Larval Density

**DOI:** 10.3390/insects12110956

**Published:** 2021-10-20

**Authors:** Idriss Hamidou Leyo, Zakari Moussa Ousmane, Gregoire Noël, Frédéric Francis, Rudy Caparros Megido

**Affiliations:** 1Ecole Doctorale Science de la Vie et de Terre EDSVT, Faculté d’Agronomie, Université Abdou Moumouni de Niamey, Niamey BP 10960, Niger; o.zakari@gmail.com; 2Entomologie Fonctionnelle et Évolutive, Terra, Gembloux Agro-Bio Tech, Liège-Université, Passage des Déportés 2, 5030 Gembloux, Belgium; gregoire.noel@uliege.be (G.N.); frederic.francis@uliege.be (F.F.); r.caparros@uliege.be (R.C.M.)

**Keywords:** egg load, substrates reduction, *Musca domestica*

## Abstract

**Simple Summary:**

The amount of waste produced by the population creates general health problems in terms of public health and hygiene. In recent years the common housefly (*Musca domestica* L. 1758) has been widely used in the treatment of organic wastes. This study aims to assess the effect of egg loading of the common housefly on maggot development and waste reduction. To do this, several housefly egg loads were incubated on three different substrates. This study indicated that larval biomass, larval number, the survival rate from egg hatching until the last larval instar and substrate rate reduction of *Musca domestica* are affected by the egg load, substrate type and their interaction. It was found that under the same nutritional conditions, the yield of housefly larvae, the number of larvae and the reduction of substrates increased with increasing egg load.

**Abstract:**

The amount of waste produced by the population creates general health problems in terms of public health and hygiene. In recent years the common housefly (*Musca domestica* L. 1758; Dipteran: Muscidae) has been widely used in the treatment of organic wastes. This study aims to assess the effect of egg loading of the common housefly on maggot development and waste reduction. Housefly larvae were reared at four egg loads (1.25, 2.5, 5, 10 mg) under three different diets (wheat bran, millet bran, cow dung). Two-factor ANOVA (α = 0.05) was used to test the effect of two fixed factors (egg load and substrate) on larval biomass, the survival rate from egg hatching until the last larval instar, number of larvae and substrate reduction rate. The comparison of means based on Duncan’s test was performed to compare the means of the different variables measured. Principal component analysis (PCA) was used to determine the relationship between the measured variables (larval biomass, the survival rate from egg hatching until the last larval instar, number of larvae, and substrate reduction rate) on the discrimination of the egg load factor. The results showed that under the same nutritional conditions, the yield of housefly larvae, the number of larvae and the reduction of substrates increased with increasing egg load. Indeed, at each of three substrates, the rearing egg load of 10 mg resulted in the maximum larval yield, maximum number of larvae, and maximum substrate reduction rate. At this optimum load, wheat bran generated greater biomass, greater number of larvae and greater reduction of substrate compared to millet bran and cow dung. The egg load as a whole had no effect on the survival rate from egg hatching until the last larval instar, unlike substrate type. The high egg load for the survival rate (from egg hatching until the last larval instar) for millet bran was 1.25 while there was no difference for the other two substrates. These results can help to make the waste treatment process efficient with the subsequent production of a large larval biomass that can serve as added value in animal feed. The egg load of 10 mg and the wheat bran were superior respectively to the other egg load and substrates type for all parameters tested excepted for the survival rate (from egg hatching until the last larval instar). Ours study indicated that larval biomass, larval number, egg viability and substrate rate reduction of *Musca domestica* are affected by the egg load, substrate type and their interaction.

## 1. Introduction

Since the start of the 19th century, the global urban population has grown from 220 million to over 4 billion. Consequently, the amount of waste produced has multiplied 20-fold, bringing the amount of household waste to 2.01 billion ton per year today and it is expected to grow to 3.40 billion ton by 2050 [1]. For example, during 2016, 174 million ton of household waste was generated in sub-Saharan Africa (currently 400 million inhabitants), representing about 0.46 kg of waste per capita per day [2,3]. In particular, Niger is experiencing strong demographic pressure, with a population estimated at 25 million inhabitants in 2020 [4] and a population growth rate of 3.87%, which is the highest globally. The capital of Niger, Niamey, is the most populous city in Niger, with 1,336,000 inhabitants, of which approximately 43% practiced agriculture in 2008. This strong growth of the urban community of Niamey (UCN) has led to the emergence of agriculture in urban and peri-urban areas. In 2015, animal herds in Niamey, capital of Niger, were estimated at around 105,212 TLU (tropical livestock units) [5].

Urbanization of the UCN increases the amount of waste generated each year. The citizens of UCN produce 1000 tons of waste generated daily [6]. This quantity of waste produced by the population creates widespread concerns in terms of public health and hygiene. Particular issues include restricted flow of runoff water, blocked gutters, and blocked wastewater ducts, including solid waste ducts. The accumulation of this waste creates favorable conditions for the development of vectors of diseases, including malaria and cholera. In parallel, pestilential odors arise, which considerably hamper the use of public spaces. Consequently, the aesthetic appeal of the city is masked by pollution, piles of rubbish extending from dumpsites and obstructing streets [2,3]. The governments should develop solutions to manage these various complex challenges.

Typical methods used for waste treatment include incineration, composting, and anaerobic digestion [7]. However, several studies, focus has been placed on using insects in the bioconversion of waste as a solution for recycling waste and reducing food waste [8,9,10]. This alternative approach allows organic waste to be recycled, and parallel converting several tons of organic waste to protein-rich larval biomass [11]. This larval biomass can be used in both human food and animal feed. Maggot meal could partially or totally replace fishmeal in animal feed, particularly poultry feed, with no negative effects [9,11,12,13,14,15,16,17,18,19,20]. The recycling of waste and the production of protein are two of many benefits in using insects for waste treatment. The use of insects also generates a residue that is rich in minerals, which can be valorized as a quality bio-fertilizer to improve soil quality, enhance crop yield, and reduce the use of chemical fertilizers [21,22,23,24,25]. Furthermore, this residue can be used to produce secondary industrial compounds (e.g., biofuel, lubricants, pharmaceuticals, dyes) [26].

Several fly species are used for the organic recycling of waste, including *Hermetia illucens* (L. 1758), *Musca domestica* (L. 1758), *Musca autumnalis* (De Geer 1776), *Lucilia sericata* (Meigen 1826) and *Sacrophaga carnaria* (L. 1758) [14,27,28,29,30,31]. *M. domestica*, commonly known as housefly, is the species of fly that is most widely used in organic waste recycling. This cosmopolitan fly has the advantage of being able to grow in a broad spectrum of wastes and has a shorter development cycle than other species in Africa [32,33,34,35,36,37,38,39,40,41,42]. There are various methods used in maggot production systems. The first production system of housefly maggot is based on the exposure of substrates (i.e., pig manure, poultry manure, cattle manure, slaughterhouse waste) placed in a container (bucket, calabash, pot...), to attract adults for oviposition. A few days later, the substrate is sifted to extract mature maggots, which are either sun dried or given fresh to the animals. On the other hand, the maggot production system is based on the rearing of adult fly colonies in cages with various sizes. These fly colonies are placed in production rooms maintained under given conditions of temperature, relative humidity, and photoperiod. For more information on the maggot production method, the literature review of [43] gives us a better global overview on the fly maggot production method.

In order to take advantage and to harness the capacity of the housefly to reduce waste material into high quality animal protein, this species needs to be mass-reared or farmed. The housefly intensive farming is not very different to intensive farming of livestock. Many factors are limiting the mass-rearing of housefly and often referred to as density-dependent processes, temperature, and substrate [44,45,46,47]. While a few studies have looked on different larval densities on various substrates [27,37,44,48], there are no published studies that investigated simultaneously the effect of several larval density of housefly and several substrates on the biological parameters of housefly.

It has been reported that housefly larvae have a high crude protein content (40 to 60% DM) and lipid content (9 to 25% DM) [49] and lysine and methionine (the two most limiting essential amino acids) were found to be higher in maggot meal than in fish meal and other conventional protein sources [50]. It is also show that the fatty acid profile of housefly larvae is suitable for broiler growth [51]. It was reported that the bioconversion of waste with housefly larvae offered a digested residue which can be used as a quality biofertilizer to improve soil quality, increase crop yields and reduce the use of chemical fertilizers [43]. In addition, this digested residue can be sold and thus create revenue [52]. All these advantages linked to the waste management using the housefly may well benefit poor countries as Niger which has low financial means and a population with strong demographic growth generating more and more important quantities of waste and where poultry farming is an activity practiced by more than 80% of the population in Niger [53].

The current study aims to evaluate the impact of different eggs load of *M. domestica* on maggot development and waste reduction, using three different substrates for larval development. The hypothesis is that waste reduction and larval yield increase with egg load regardless of the larval development substrate used.

## 2. Materials and Methods

### 2.1. House Fly Colony

Insects used in this experiment originated from the mass breeding of *M. domestica* at the Faculty of Agronomy in Abdou Moumouni University (Niamey, Niger). Five breeding cages (75 × 75 × 115 cm BugDorm, Mega View Science, Taichung, Taiwan) were used to maintain breeding. Each cage contained 25,000 reared *M. domestica* pupae corresponding to a storage density of approximately 2.8 cm^3^ per fly [40]. Cotton soaked in a mixture of powdered milk and granulated sugar (ratio 1:1), and sponges soaked in sweet water placed in plastic containers were used as food for the adults. The cages were placed in a room with photoperiod of 12 h light and 12 h dark (12:12 L:D), temperature of 25 ± 2 °C, and relative humidity (RH) of 60–70% [40,54].

### 2.2. Effect of Egg Load on Larval Productivity

Five days after adult emergence, plastic containers (83 mm diameter and 3 cm height) containing a mixture of water, wheat bran, and granulated sugar (2:1; 70% moisture) covered with filter paper (grade: 50, circular, porosity: 2.7 µm; Whatman, La Chapelle-sur-Erdre, France) were placed in colony cages as spawning medium [40]. Flies were allowed to lay eggs for 8 h (8.00 to 16.00 h). Filter paper was used just to prevent flies from laying eggs on the mixture of fermented wheat bran and granulated sugar as spawning medium and allow easy collection of eggs. The eggs were weighed on a balance (0.001 g, Sartorius, Goettingen, Germany) using a fine brush in the breeding room where humidity is kept relatively high (60–70% RH) to avoid desiccation of the eggs and water loss from larval medium [55]. Four different egg loads (i.e., 1.25, 2.5, 5 and 10 mg), similar to egg load used on cattle manure by [37], were tested on three different substrates: millet bran, wheat bran, and cow dung. Millet bran and wheat bran are the less expensive or even free substrate from household waste and these substrates are the most accessible and available products in UCN. Cow dung is the most common materials found in UCN. Twenty grams of dry substrate was used and the humidity of the different substrates was kept at 70% by adding tap water. Each egg load was incorporated in the substrate per replicate (*n* = 3). All maggots were reared in plastic containers (17.20 × 11.50 × 6.00 cm, AVA, Temse, Belgium) covered with a transparent lid with a fine tissue for ventilation. Containers were randomly arranged at half-height on a board. The experiment was implemented under the same environmental conditions as the rearing of adults (26 °C ± 2 °C; 60–70% RH and 12:12 L:D).

### 2.3. Evaluation

Five days after placing the eggs in the different substrates, the larval biomass, larval number, egg viability (EV), and final substrates (substrate residue after larval extraction) were measured by replicate. Larval biomass was weighed on a balance (0.001 g, Sartorius, Goettingen, Germany) and the larval number (LN) was counted per replicate. To calculate egg viability (EV), the number of eggs in each egg load was counted beforehand: the eggs were placed in petri dishes (90 mm) half-filled with water and sifted using filter paper to separate them for counting. Survival larval rate until the last larval instar (SR) was calculated by dividing the number of larvae by the number of eggs corresponding to each egg load incorporated per replicate. The final substrate of each replicate was dried in the oven at 70° for 24 h and weighed. The reduction in substrate rate (SRR) of each replicate was calculated following the method described by [56] used for *Hermetia illucens*: substrate reduction rate (SRR) = [(distributed substrate (g) − residual substrate (g))/distributed substrate (g)] × 100.

### 2.4. Statistical Analyses

All statistical analyses and graphics were performed on R version 4.0.3 environment [57]. Two-factor ANOVA (α = 0.05) was used to test the effect of two fixed factors (egg load and substrate) on larval biomass, egg viability, number of larvae and substrate reduction rate. The comparison of means based on Duncan’s test was performed to compare the means of the different variables measured (larval biomass, egg viability, number of larvae, and substrate reduction rate) on the different egg loads according to the different substrates as well as to compare the means of the different substrates of these same variables according to each egg load. Principal component analysis (PCA) was used to determine the relationship between the measured variables (larval biomass, egg viability, number of larvae, and substrate reduction rate) on the discrimination of the egg load factor (egg load). The following R packages *agricolae* and *FactoMineR* were used during analyses [58,59].

## 3. Results

### Effect of Substrate Type and Egg Load of Musca domestica on Maggot Development and Waste Reduction

Two-way analysis of variance (Table 1) shows that larval biomass, larval number, and substrates rate reduction were strongly influenced by the substrate type and egg load as well as the interaction of the two factors. Overall, egg load has no effect on the survival rate (from egg hatching until the last larval instar; SR). However, substrate type and its interaction with egg load have a significant effect on egg viability.

The ANOVA results (Table 2—vertical comparison) show that the larval biomass is influenced by the egg loads for each substrate. For all three substrates, the highest biomass is obtained with an egg load of 10 mg. At this high egg load (10 mg), wheat bran generated a higher larval biomass than the other two substrates. It was found that egg viability was influenced by egg load only for the millet bran substrate, whose optimal average was obtained at the egg load of 1.25 mg (95.00 ± 5.00%). The number of larvae measured was also significantly influenced by the egg loads for each level of substrate. The high average larval number was obtained with the egg load of 10 mg (wheat bran =156.66 ± 1.52, cow dung = 155.00 ± 3.46, millet bran = 150.00 ± 2.00). With respect to substrate reduction by maggots, the egg load had a significant effect for each substrate as the 10 mg egg load provided the highest reduction rate with an average of 26.00 ± 1.00, 24.33 ± 0.57, and 22.33 ± 0.58 for wheat bran, millet bran, and cow dung, respectively.

Larval biomass was significantly influenced by substrates for each egg load considered (Table 2—horizontal comparison). For each corresponding egg load, wheat bran provided the highest larval biomass with an average of 488.33 ± 9.60 mg, 855.33 ± 49.90 mg, 1712.33 ± 184.85 mg, and 3240.00 ± 76.26 mg for the 1.25, 2.5, 5, and 10 mg egg load, respectively. SR was not significantly (*p* = 0.272) influenced by the substrates for the 1.25 egg load in contrast to the other egg loads. The 10 mg egg load had recorded the highest SR that was essentially identical to the wheat bran and cow dung with an average of 98.00 ± 1.00 and 96.66 ± 2.30%, respectively. While the highest SR for millet bran (95.00 ± 5.00%) was obtained with the egg load of 1.25 mg. Regarding the number of larvae, there was no significant difference (*p* = 0.422) between the substrates for the 1.25 mg egg load. Nevertheless, the 10 mg egg load resulted in a significantly different average number of larvae between the three substrates. At this highest egg load (10 mg), the average number of larvae was 156.66 ± 1.52, 155.00 ± 3.456, and 150.00 ± 2.00, respectively, for wheat bran, cow dung, and millet bran. For the ability of maggots to reduce their feeding substrates, there was a significant difference between substrates for each egg load considered. At the level of each egg load, wheat bran rerecorded the highest reduction rate with an average of 05.33 ± 0.57, 15.00 ± 1.00, 23.00 ± 1.02, and 26.00 ± 1.00% for the 1.25, 2.5, 5, and 10 mg egg load, respectively.

The correlation circle obtained from the PCA based on the first two axes explains the relationship between the measured variables (Figure 1A) and shows the discrimination of egg load (Figure 1B) according to the measured variables. The first two dimensions explain 95.7% of the total information, with 70.3% for dimension 1 and 25.4% for dimension 2 (Figure 1A). SR is positively correlated with dimension 2 (correlation coefficient greater than 0.5), while larval biomass (Biom), number of larvae (LN) and substrate reduction (SRR) are positively correlated with dimension 1 (correlation coefficient greater than 0.5) and are also positively correlated with each other (Figure 1A). However, SR is an independent variable from the other three variables (Figure 1A). The dimension 2 (Figure 1B) discriminates well the egg loads according to its increasing order. The four egg loads were clearly differentiated from each other, the 1.25 mg and 2.5 mg loads are significantly close to each other, while the 10 mg load is well distinguished from all other egg loads (Figure 1B). The 5 mg load is closer to the 2.5 mg and 10 mg loads than to the 1.25 mg load.

## 4. Discussion

The treatment of organic waste with fly larvae is a promising technology to reduce and recycle food waste responsible for unsanitary conditions encountered in urban areas into useful products. Fly larvae are used for biodegradation of multiple wastes, such as food/restaurant waste, meat processing waste, slaughterhouse waste, municipal garbage, agricultural waste, and market waste [25,31,40,50,60,61,62]. Moreover, bioconversion of waste with housefly larvae may result in significant production of feed ingredient as housefly larvae have a suitable nutritional composition and can serve as a replacement for fish meal and other protein sources used in poultry nutrition [18,32,33,34,35,36,37,38,39,40,41,42,63]. Insects are sensitive to many abiotic and biotic factors [64,65]. In particular, the life cycle parameters of *M. domestica* larvae are influenced by temperature, humidity, diet, and rearing density. The current study showed that the larval biomass and number of larvae on three rearing substrates were significantly influenced by egg load and substrate type. After 5 days of growth, larval biomass increased as a function of egg load for all substrates. The optimal biomass was obtained with an egg load of 10 mg on wheat bran. Under the same nutritional conditions, the yield of housefly larvae and number of larvae increased with increasing breeding density. However, when breeding density continued to increase, yield tended to become stable [8,37,40,66]. At relatively high breeding densities, larvae cannot consume enough nutrients for growth and development, resulting in small individuals with incomplete development, or even death, leading to the stabilization or, even, decline in yield [40,67,68]. In particular, an egg load greater than 16 eggs/g of manure could lead to overpopulation, which would reduce larval biomass, due to insufficient nutrients being available for larval development [37]. An optimum larval density of 0.225 g CSMA (Chemical Specialties Manufacturers’ Association) larval medium per egg was described in one study [69], but since these various densities on various substrates were explored [27,44,48] with stocking densities mostly expressed per unit of feed source. For example, densities evaluated ranged from one larva per gram of poultry manure or cattle manure [27,37,44] to 16 larva per gram [37,48]. However, larvae production was found to be closely correlated (R > 0.958) with stocking densities [27]. The optimum stocking density to obtain a significant quantity of pupae is 3 g of eggs per 4 kg of fresh poultry manure, while the density of 2 g of eggs per 4 kg of fresh manure results in individually larger larvae [48]. A later study [44] summarized the influence of temperature and density (uncrowded = 1 larva/g manure; moderately crowded = 2.5 larvae/g manure; crowded = 5 larvae/g manure) with fastest larval development at 32 °C. The most variation in larval size was observed in uncrowded larvae, and survival rates were the best at 23 °C uncrowded. Pupal mass was highest for uncrowded larvae.

In the current study, wheat bran produced a higher larval biomass compared to cow dung and millet bran, regardless of egg load. These results were attributed to wheat bran having a loose, less consistent, and more aerated structure compared to cow dung and millet bran, which was more compact with high moisture loss. Several studies stated that maggot yield varies greatly with the characteristics of the substrate used, including odor, texture, decomposition rate, moisture holding capacity, and chemical composition [37,38,70,71,72].

In the present study survival rate from egg hatching until the last larval instar (SR) was influenced by substrate type. This difference could be explained by the low moisture holding capacity observed on millet bran. The different types of diets and their moisture content are factors that influence the development of M. domestica [73,74,75]. Low humidity causes water loss through the membrane of insect eggs, resulting in the desiccation and failure of embryos to hatch [54]. Females of many fly species, including the housefly, select spawning sites very carefully, preferentially laying eggs on surface structures with substrate characteristics that maximize offspring survival [76,77]. Spreading eggs over a substrate surface increases the risk that some eggs are placed in a location with favorable properties, reducing competition between larvae early in their development. Simple dispersal of eggs on substrate can increase larval survival, allowing higher larval biomass to be produced [71].

Substrate rate reduction (SRR) increased with increasing breeding density for all three substrates and was more pronounced on wheat bran. The optimum load of eggs for the biodegradation of different types of waste used was 10 mg/20 g substrate. This result was similar to that obtained by [37], who recorded an increasing rate of substrate reduction of 4.7%, 11.2%, 20.6%, 25.3%, and 26.5% for egg loads of 1, 2, 4, 8, and 16 eggs/g manure, respectively. The authors concluded that higher egg loading enhanced biodegradation. [78] estimated that the optimum quantity of M. domestica eggs for complete biodegradation was 0.4 mL (4400 eggs)/kg for a mixture of manure with sawdust and 1 mL (11,000 eggs)/kg for fresh manure. In comparison, [40] stated that the optimum egg load for complete biodegradation was 1.5 g (24,000 eggs)/kg for a mixture of food waste with wheat bran. However, the number of larvae converting a certain amount of food waste increased with increasing breeding density, leading to a relative lack of food for housefly larvae, forcing them to feed on foods with low nutritional value, such as vegetable protein and crude fiber to survive [66]. Therefore, breeding density contributes to the conversion rate of substrate. However, when breeding density noticeably increased, the average rate of substrate reduction decreased [67]. This phenomenon might be attributed to the excessive feeding density, which causes body weight to decline and hinders development in housefly larvae. This phenomenon reduces the feeding capacity and activity of larvae [52], and negatively affects larval yield and substrate rate reduction. Consequently, a moderate increase in breeding density could promote the conversion efficiency of housefly larvae to substrate.

## 5. Conclusions

This study contributes to the efficient use of fly maggots in waste treatment. Many advantages are linked to the waste management using the housefly which may well benefit poor African countries. To be efficient, larval density used to treat the waste should be controlled to increase larval yield and achieve effective and efficient waste reduction. Under the same nutritional conditions, the yield of housefly larvae, the number of larvae and the reduction of substrates increased with increasing egg load. At each of three substrates, the rearing egg load of 10 mg resulted in the maximum larval yield, maximum number of larvae, and maximum substrate reduction rate. Larval density should be controlled to increase larval yield and obtain effective and efficient reduction in waste. Optimal larval density enhances the performance (efficiency and effectiveness) of biodegradation facilities and larval biomass production. Perhaps higher egg loads would have negatively impacted these parameters. This study probably did not reach the maximum production densities so it would be interesting to develop models to predict more increasing biomass depending on the substrates used.

## Figures and Tables

**Figure 1 insects-12-00956-f001:**
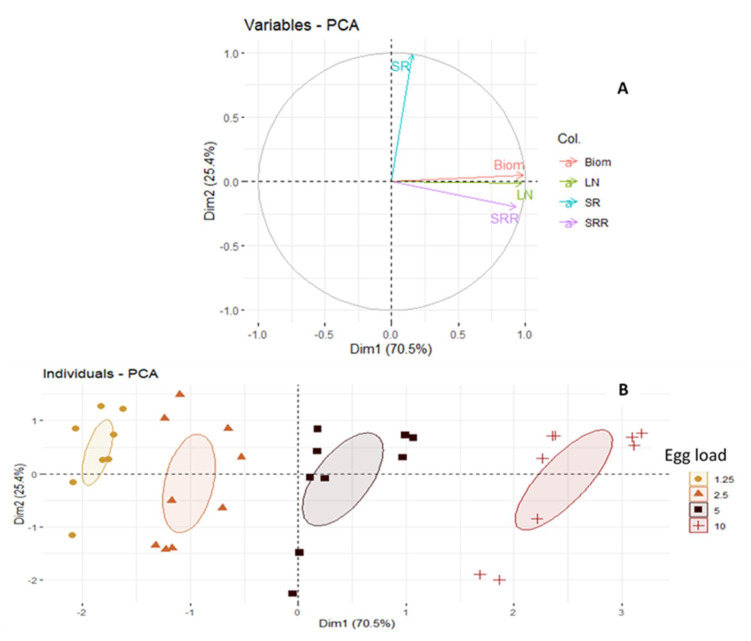
(**A**) Circle of correlation of variables. (**B**) Graphical representation of individuals as a Figure. Biom: biomass; LN: larval number; SR: survival rate; SRR: substrate reduction rate.

**Table 1 insects-12-00956-t001:** Summary of the two-factor ANOVA test on the influence of egg load, substrate, and their interaction on the observed parameters.

Parameters	Factors	Df	F Value	*p*-value
Biomass	Egg load	3	1278.95	*p* < 0.001
	Substrates	2	109.36	*p* < 0.001
	Egg load × Substrates	6	21.29	*p* < 0.001
Survival rate	Egg load	3	0.63	0.600
	Substrates	2	16.31	*p* < 0.001
	Egg load × Substrates	6	4.194	0.005
Larval number	Egg load	3	3496.44	*p* < 0.001
	Substrates	2	20.05	*p* < 0.001
	Egg load × Substrates	6	3.16	0.019
Substrate rate reduction	Egg load	3	453.10	*p* < 0.001
	Substrates	2	58.38	*p* < 0.001
	Egg load × Substrates	6	3.03	0.023

The *p*-values in italic indicate significant tests (*p* < 0.05).

**Table 2 insects-12-00956-t002:** ANOVA of biological parameters inside different substrates as a function of egg load and between the different substrates.

Parameters	Egg Lad (mg)	Millet Bran	Wheat Bran	Cow Dung	Statistical Analysis
Biomass (mg)	1.25	412.67 ± 11.93 c (b)	488.33 ± 9.60 d (b)	396.33 ± 23.02 d (a)	*p* < 0.001; Df = 2; F = 28.35
	2.5	506.00 ± 95.39 c (c)	855.33 ± 49.90 c (b)	726.33 ± 17.09 c (a)	*p* = 0.001; Df = 2; F = 23.63
	5	1267.00 ± 146.24 b (b)	1712.33 ± 184.85 b (a)	1281.33 ± 1.52 b (b)	*p* = 0.011; Df = 2; F = 10.37
	10	2194.67 ± 61.04 a (c)	3240.00 ± 76.21 a (b)	2458.66 ± 63.10 a (a)	*p* < 0.001; Df = 2; F = 196.80
	Statistical analysis	*p* < 0.001; Df = 3; F = 238.70	*p* < 0.001; Df = 3; F = 421.90	*p* < 0.001; Df = 3; F = 2047.00	
Survival rate (%)	1.25	95.00 ± 5.00 a (a)	93.33 ± 5.77 a (a)	85.00 ± 10.00 a (a)	*p* = 0.272; Df = 2; F = 1.63
	2.5	75.00 ± 1.00 b (b)	91.33 ± 7.63 a (a)	94.66 ± 10.40 a (a)	*p* = 0.037; Df = 2; F = 5.98
	5	77.33 ± 10.69 b (b)	97.00 ± 1.73 a (a)	93.66 ± 4.50 a (a)	*p* = 0.025; Df = 2; F = 7.24
	10	75.33 ± 6.65 b (b)	98.00 ± 1.00 a (a)	96.66 ± 2.30 a (a)	*p* < 0.001; Df = 2; F = 28.74
	Statistical analysis	*p* = 0.018; Df = 3; F = 6.03	*p* = 0.363; Df = 3; F = 1.22	*P* = 0.322; Df = 3; F = 1.36	
Larval number	1.25	19.00 ± 1.00 d (a)	19.66 ± 0.57 d (a)	20.00 ± 1.00 d (a)	*p* = 0.422 Df = 2; F = 1.00
	2.5	30.66 ± 1.51 c (b)	37.00 ± 2.00 c (a)	38.66 ± 1.52 c (a)	*p* = 0.001; Df = 2; F = 20.91
	5	63.33 ± 8.50 b (b)	78.01 ± 2.05 b (a)	75.33 ± 2.51 b (a)	*p* = 0.030; Df = 2; F = 6.64
	10	150.00 ± 2.00 a (b)	156.66 ± 1.52 a (a)	155.00 ± 3.46 a (a)	*p* = 0.038; Df = 2; F = 5.99
	Statistical analysis	*p* < 0.001; Df = 3; F = 534.90	*p* < 0.001; Df = 3; F = 4187.00	*p* < 0.001; Df = 3; F = 1978.00	
Substrate rate reduction (%)	1.25	05 ± 0.01 d (a)	05.33 ± 0.57 d (b)	02.00 ± 1.01 d (a)	*p* = 0.006; Df = 2; F = 13.00
	2.5	12.66 ± 1.52 c (a)	15.00 ± 1.00 c (b)	07.66 ± 1.52 c (a)	*p* = 0.001; Df = 2; F = 22.29
	5	18.66 ± 2.08 b (b)	23.00 ± 1.02 b (b)	15.66 ± 0.57 b (a)	*p* = 0.005; Df = 2; F = 14.20
	10	24.33 ± 0.57 a (b)	26.00 ± 1.00 a (c)	22.33 ± 0.58 a (a)	*p* = 0.002; Df = 2; F = 18.20
	Statistical analysis	*p* < 0.001; Df = 3; F = 102.90	*p* < 0.001; Df = 3; F = 162.10	*p* < 0.001; Df = 3; F = 239.00	

Italic value indicates significant tests (*p* < 0.05) and the brackets for the letters correspond to horizontal tests. The letters are from the Duncan comparison at threshold α = 0.05.

## Data Availability

The data presented in this study are available in this article.

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
