# Peer review of "Breeding Enhancement of Musca domestica L. 1758: Egg Load as a Measure of Optimal Larval Density"

_insects, 2021, doi:10.3390/insects12110956_

Round 1
Reviewer 1 Report
Comment to Authors,
The manuscript "Breeding Enhancement of Musca Domestica L. 1758: Egg 2 Load as a Measure of Optimal Larval Density" contains some interesting and relevant findings. Certain sections require more details before publication is possible. However, the authors should be capable of resolving these issues.
Points requiring further clarification:
1) Introduction, lines 109-110 - Explain a bit more - briefly describe what you are referencing.
2) Results, lines 219-220, 233 - What is meant by "optimal"? It is mentioned several times, but not clarified until the Discussion (page 11). Clearly define "optimal" with a value or description. mg/g?
3) line 234 - the units (g) need to be included
Typos & Other:
1) Title - Breeding Enhancement of Musca Domestica (lower case d)
2) Simple Summary, lines 14 & 18 - Musca domestica must be italicized
3) line 16 - "reduction. to do this, several Housefly egg" capital T, lower case h
4) line 17 - "substrates. this study" capital T
5) line 24 - "(Musca domestica L. 1758) has been widely" also need (Diptera: Muscidae)
6) line 108 - "and photoperiodicity" photoperiod
7) line 112 - "this fly need to be mass-reared or farmed." this species (or Dipteran, organic waste consumer, etc.) needs...
8) line 121 - "lipid content (9 to 25% DM) [49] and lysine and" [49]. Lysine
9) line 130 - "countries such as Niger which has" as Niger, which
Author Response
all comments are accepted.
Reviewer 2 Report
The article addresses an interesting topic with practical application, which involves rearing of Musca domestica on three different substrates.
Fly larvae can be used to decompose organic matter (food scraps, for example), aiming at the production of fertilizers and biofuels.
The article is generally well written. Several scientific names (binomials) appear with the specific epithets starting with a capital letter, from the title to the references.
What was the time interval between hatching larvae and collecting data on survival, biomass and number of formed larvae?
After the larvae hatch, there may have been early larval mortality. Thus, I consider necessary to replace "egg viability" by "survival until the last larval instar".
Although the direct practical application of the work is the reduction of food waste through the reuse of organic matter, it lacked a comparison with works contemplating other quantitative methods for analyzing mass rearing data.
The discussion could start with the second paragraph. In the first, comments are made, among other topics, on the nutritional composition of substrates, which was not covered in depth in this work. Furthermore, line 9 on page 11 comments on stability with increasing breeding density. Was there stability in relation to the results of the present work?
In the conclusions, penultimate line: greater larval densities/ biomass?
Author Response
The time interval between hatching larvae and collecting data was 5 days.
This study probably did not achieve the maximum egg densities for production and the amount of substrate used in the present study supported the egg densities used so stability was not observed in the present study
In the conclusions, penultimate line: greater larval densities is change by biomass.